# Energy Content and Nutrient Profiles of Frequently Consumed Meals in Singapore

**DOI:** 10.3390/foods10071659

**Published:** 2021-07-19

**Authors:** Penny Liu Qing Yeo, Xinyan Bi, Michelle Ting Yun Yeo, Christiani Jeyakumar Henry

**Affiliations:** 1Clinical Nutrition Research Centre (CNRC), Singapore Institute of Food and Biotechnology Innovation (SIFBI), Agency for Science, Technology and Research (A*STAR), 14 Medical Drive, Singapore 117599, Singapore; penny_yeo@sifbi.a-star.edu.sg (P.L.Q.Y.); bi_xinyan@sifbi.a-star.edu.sg (X.B.); michelle_yeo@sifbi.a-star.edu.sg (M.T.Y.Y.); 2Department of Biochemistry, Yong Loo Lin School of Medicine, National University of Singapore, Singapore 117599, Singapore

**Keywords:** energy content, nutrient profiles, ethnic cuisines, calorie answer, inductively coupled plasma mass spectroscopy (ICP-MS)

## Abstract

Singapore is a multi-ethnic country with a great variety of traditional ethnic cuisines. In this modern society where there is an increasing prevalence of obesity, it is important to know the nutritional content and energy density of our foods. However, there have been little data on the nutritional content of our local foods. The energy density and nutrient content of 45 commonly consumed meals by three ethnic groups in Singapore (Chinese, Malay, and Indian) were assessed in this study. Chinese, Malay, and Indian cuisines had an average energy density of 661, 652, and 723 kJ/100 g, respectively. Moreover, the macronutrient content is different between the different ethnic groups. Compared to Chinese and Malay cuisines, Indian cuisine contained lower protein but higher fat and carbohydrate content (*p* = 0.03). From the mineral analysis of the ethnic foods, we found out that Chinese cuisines contain significantly higher sodium (average of 238 mg/100 g) than Malay cuisines (*p* = 0.006) and Indian cuisines (*p* = 0.03). Knowing the caloric density and nutrition content of local ethnic foods may aid hawkers and government officials in developing healthier options to tackle Singapore’s obesity epidemic.

## 1. Introduction

In modern society, there has been an exponential increase in obesity due to the changes in diet and “nutrition transition”. This consists of the increase in consumption of fat, meat, added sugars, and bigger portion sizes, as well as the decrease in physical activity [1]. This global epidemic is particularly visible in affluent nations, such as Singapore, where the incidence of obesity is steadily increasing [2]. The proportion of obese and overweight adults aged 18 and 69 years was 8.7% and 36.2%, respectively, in 2017, as compared to 8.6% and 34.3%, respectively, in 2013 [3].

Singapore is a microcosm of Asia where three broad ethnicities corresponding to the major population centers in Asia are present, namely the Chinese (East Asians), the Malays (Southeast Asians), and the Indians (South Asians) [4]. Street food and foods created by tiny local businesses, known as hawker centers or Kopitiam in the local language, are popular venues for locals to have breakfast, lunch, and supper, as they are in many other Southeast Asian nations. From the Food Forward Trends Report Singapore in 2019 [5], the proportion of Singaporeans eating out is higher (61%) compared to the past years. Hawker center and fast food are listed as one of the top three eating out options in Singapore.

However, there is insufficient information on the caloric content of local ethnic foods, with the majority of the information dating back many years and based on bomb calorimetry and Atwater conversion factors [6]. We recently employed a new instrument called the Calorie Answer^TM^ to determine the calorie content of various foods [7,8]. Our results showed that this near-infrared spectroscopy (NIR) is rapid and reliable for a diverse range of foods [7].

Obesity is recognized to be a major risk factor for a variety of ailments, including cancer, heart disease, and diabetes mellitus, which are among the top 10 diseases afflicting Singaporeans [9,10]. Diet-related disorders, such as high blood pressure, are also affecting Singaporeans, which is linked to a higher salt consumption [11]. The intake of sodium by Singaporeans has increased over the years to 9 g, which has exceeded the recommended daily intake of 5 g [12]. In addition, minerals are needed by the body to ensure the internal systems function efficiently. Therefore, the dietary mineral intake must be monitored to maintain physical health. However, there are limited data on the mineral composition of local ethnic cuisines in Singapore.

Data on the nutritional content of these local ethnic foods are critical for determining nutrient intake, giving dietary recommendations, and avoiding illnesses. Therefore, the objective of the study is to analyze the energy density, macronutrient content (fat, protein, and carbohydrates), and mineral content (Na^23^, Mg^24^, Al^27^, K^39^, Ca^40^, Mn^55^, Fe^56^, Cu^63^, Zn^66^) of 45 local ethnic cuisines commonly consumed in Singapore.

## 2. Materials and Methods

### 2.1. Sample Preparation

Local food samples were purchased from a local food court (Kopitiam, Singapore). There were, in total, 45 local food samples from different ethnic groups that were used for analysis: Chinese food (*n* = 15), Malay food (*n* = 15), and Indian food (*n* =15). To achieve a smooth, consistent texture, all of the samples were homogenized in a kitchen blender (BL 480, Kenwood, United Kingdom). The samples were then put into 6 tubes, 3 for Calorie Answer analysis, and 3 for inductively coupled plasma mass spectroscopy (ICP-MS) analysis. For the ICP-MS analysis, the samples were kept in a −20 °C freezer. The samples were analyzed using the procedure and techniques described previously. The foods are described in Appendix A.

### 2.2. Standards and Reagents

Analytical and trace-metal grade reagents were utilized throughout. The Mili-Q IQ-7000 (Merck KGaA, Darmstadt, Germany) water purification system was used to provide high quality deionized water (resistivity > 18.2 M). Prior to use, all glassware and plasticware were cleaned by soaking for 48 h in a 10% (*v*/*v*) HNO3 solution, then rinsing with ultrapure water and drying. The dilution of 10 µg/mL (Bi^209^, Tb^159^, In^115^, Y^89^, Ge^74^, Ge^72^, Sc^45^, Li^6^) Internal Standard Mix solution (Agilent Technologies, Santa Clara, CA, USA) was required for the internal standard solution. The calibration curves were made using a 10 µg/mL and 100 µg/mL custom mix multi-element ICP-MS standard (HPS, North Charleston, SC, USA), as well as a 1000 µg/mL mineral element ICP-MS standard solution (HPS, North Charleston, SC, USA) (ICP-AM-17 Mineral Calibration Standard, HPS, North Charleston, SC, USA).

### 2.3. NIR Spectroscopy and Analysis

The homogenized samples were then put in cylindrical sample cells (with an internal diameter of 50 mm and a depth of 10 mm for solid samples) and tested in triplicates. For all samples, the Prepared Food settings were used. For solid samples, the reflectance mode was applied, and the reference reflectance data were analyzed using a calcium-carbonate-filled cell. Calorie Answer^TM^ (CA-HM, JWP, Hirakawa, Japan) was used to collect near-infrared (NIR) spectra of homogenized samples throughout a wavelength range of 1100–2200 nm, with a resolution of 7.5 nm and a data interval of 2 nm. A halogen lamp served as the radiation source, while an acousto-optic tunable filter (ATOF) served as a wavelength sensor and light-receiving sensors served as light detectors. To increase accuracy, the integrated computer program (CA-HM Measurement Application Software, JWP, Hirakawa, Japan) was programed to scan each triplicated component 10 times, then averaged to obtain a mean spectrum. After converting the data to log 1/R, the calorie density for each sample was determined using regression formulas preprogramed in the software. Each measurement took roughly 5 min to analyze (including time for calibration). The procedures used were elaborated intensively by Lau, et al. [7].

### 2.4. ICP-MS Mineral Analysis

A CEM MARS6 Microwave Digestor System with an iPrep 12 vessels rotor (CEM, Matthews, NC, USA) was utilized to digest the local food samples. Each Teflon vessel contained 0.25 g of sample, followed by 10 mL of HNO_3_ (Fisher Chemical, Waltham, MA, USA). The mixture was heated to 210 °C in 15 min and then held for another 15 min. After cooling, sample solutions were diluted to 50 mL with 5% HNO_3_ and 0.5% HCl in decontaminated 50 mL skirted centrifuge tubes. Of these 12 vessels, ten were samples, one was blank, and one was a spiked sample from the same digestion batch. In each digesting operation, this arrangement was retained. Duplicates of each sample were digested. The 7900 ICP-MS analyzer (Agilent Technologies, Hachioji, Japan) with the Ultra High Matrix Introduction (UHMI) option was used to perform inductively coupled plasma-mass spectrometry (ICP-MS) analysis. A MicroMist nebulizer, quartz spray chamber, and quartz torch with a 2.5 mm internal diameter injector were utilized during the experiment. The interface cones used had a platinum tip. The plasma was created using high quality (99.9997%) argon (Air Liquide, Singapore, Singapore). The following were the parameters of the ICP-MS instrument: 1550 W RF power, 10 mm sampling depth, 0.90 L/min carrier gas. The samples, which were kept in 50/15 mL tubes, were delivered by an Agilent SP4 auto-sampler. Internal standards were used: Sc^45^, Ge^72^, Y^89^, In^115^, and Bi^209^. Using the tuning solution (1 g/L Ce, Co, Li, Mg, Tl, Y in 2% HNO_3_) for the ICP-MS (Agilent Technologies; Santa Clara, CA, USA), the instrument was set for optimal signal sensitivity and stability. Triplicates of each analysis were performed.

### 2.5. Method Validation and Statistical Analysis

The ICP-MS analytical technique was derived from an Agilent application note [13]. To validate the method, 9 of the 45 local foods were used to run a spike recovery test to measure the elements tested. The average recoveries were presented in Appendix A. Excellent spike recoveries were achieved, with most elements being 95–105% recovered. These foods were ranked from the highest to the lowest energy content (per 100 g). The mean and standard deviation (SD) are used to express the results.

The values of Ca^44^ were converted to Ca^40^ using the atomic abundance of the Ca element. The formula is as follows:$$Ca^{40}\;mineral\;concentration\;(ppm)=\frac{Ca^{44}\;mineral\;concentration\;\left(ppm\right)\times 96.94\%}{2.09\%}$$

## 3. Results

The energy density and the macronutrient content of the selected foods were determined and reported for 100 g edible portion, as presented in Table 1 (Chinese cuisine), Table 2 (Malay cuisine), and Table 3 (Indian cuisine).

The foods were listed from the highest to the lowest energy density for each ethnic group. The average energy density of Chinese, Malay, and Indian cuisines was 661, 652, and 723 kJ/100 g, respectively, as in Figure 1a. For Chinese cuisine, steamed white chicken rice had the highest energy density (789 kJ/100 g), while dumpling you mian had the lowest energy density (496 kJ/100 g) (Table 1). For Malay cuisine, ayam penyet had the highest energy density (849 kJ/100 g), while mee rebus had the lowest energy density (472 kJ/100 g) (Table 2). For Indian cuisine, egg prata with chicken curry had the highest energy density (782 kJ/100 g), while vegetarian set meal (biryani rice) had the lowest energy density (521 kJ/100 g) (Table 3). Figure 1b,c show that the macronutrient compositions and the mineral contents of local ethnic foods were remarkably different between different ethnic groups.

Appendix A show the mineral distributions of the foods consumed by different ethnic groups. Our results suggested that local ethnic foods are high in Na, K, and Ca. Appendix A shows that, for Chinese cuisine, economical mee goreng (per portion, same below) had the highest amount of sodium (1575 mg), and laska had the highest amount of magnesium (90 mg), potassium (251 mg), calcium (1236 mg), manganese (1.15 mg), and iron (2.89 mg). Appendix A shows that, for Malay cuisine, mee rebus had the highest amount of sodium (1170 mg), goreng pisang had the highest amount of magnesium (94.5 mg) and manganese (2.07 mg), nasi ambang chicken had the highest amount of potassium (574 mg) and iron (3.78 mg), and tahu goreng set had the highest amount of calcium (1129 mg). Appendix A shows that, for Indian cuisine, naan set had the highest amount of sodium (1219 mg), chapati set with potato marsala had the highest amount of magnesium (99.5 mg) and potassium (729 mg), vegetable biryani had the highest amount of calcium (677 mg), and poori set had the highest amount of manganese (2.37 mg) and iron (5.21 mg). Variations in local ethnic meals and recipes are assumed to be related to a variety of factors, including processing and farming conditions, as well as varied ingredients and cooking methods. As shown in Figure 1c, Chinese cuisine has a relatively high sodium content compared to other ethnic cuisines (*p* < 0.05). This may be due to Chinese cuisines using seasonings, such as soya sauce and monosodium glutamate (MSG), to season their food.

## 4. Discussion

Local ethnic foods are the main meals consumed in Singapore, and with more Singaporeans eating out, this may contribute to higher energy intake and higher sodium intake, which increases the risk of obesity and associated diseases. Since the research relating to diet and health is constantly evolving, appropriate dietary guidelines have been implemented to help Singaporeans adopt healthier food eating habits. The dietary guidelines include having a varied diet, and the foods chosen should be low in fat, especially saturated fat, low in salt, and low in sugar, replacing refined grains with whole grains, eating more fruit and vegetables each day. In recent years, there is also an increasing culture of dining out among Singaporeans from a recent 2018 Nielsen survey. The percentage of Singaporeans eating out is constantly increasing, from 51% in 2015 to 55% in 2019, on a weekly basis. Moreover, dining out meals generally deliver large portions that can lead to a substantially higher energy intake. Therefore, reducing portion size is one of the key requirements of moderating food intake. The first prerequisite for such action is the need to know the individual energy density of various local ethnic foods. In meeting this requirement, we are presenting for the first time a comprehensive list of energy density and nutrient content of various local ethnic foods.

The recommended daily allowances for the different macronutrients and micronutrients are shown in Appendix A [14]. Table 4, Table 5 and Table 6 show the macronutrient content with the %RDA of the commonly consumed Chinese, Malay, and Indian cuisines, respectively. Following the Recommended Dietary Guidelines 2003 for Adult Singaporeans, all the local ethnic foods, if eaten for all three meals, will exceed the recommended guidelines for energy contribution (%) of macronutrients to total energy intake [15]. Table 4 shows one serving of laska consumed in a day contributes to 34–43% of the caloric intake, 60–73% of protein intake, 35–45% of fat intake, and 27–35% of carbohydrates intake. From the National Nutrition Survey 2010, it was shown that 60% of Singaporeans exceeded the daily recommendation for energy and total fat [16]. This is also reflected in the National Nutrition Survey in 2018 that more Singaporeans are consuming more fat in their diet, from 31% in 2010 to 35% in 2018 [12]. It also states that protein intake was also mostly adequate, with over 80% of Singaporeans achieving the daily protein intake recommendation [12].

Local ethnic foods in Kopitiam or hawker centers can be considered “unhealthy” foods due to their high amount of sodium, MSG, and fat. Mineral contents were assessed in addition to macronutrients, since they are known to have a significant role in metabolism and tissue function. The local ethnic foods consumed in Singapore were found to be high in macro-elements, like sodium, potassium, magnesium, and calcium, but deficient in trace elements, like copper, iron, manganese, and zinc, according to this study. Table 7, Table 8 and Table 9 show the percentage of the mineral content of the different ethnic foods with comparison to the recommended intake.

Sodium is essential for the control of blood pressure and stimulation of muscles and nerves. It is an electrolyte that controls the extracellular amount of fluid in the body and is needed for hydration. Excessive consumption of dietary salt and sodium-containing substances, like monosodium glutamate (MSG), has been linked to high blood pressure, making it a risk factor for cardiovascular illnesses [17]. The daily sodium consumption requirement is 2.4 g; thus, salt intake should not exceed 6 g per day [18]. From Table 7, Table 8 and Table 9, we can see that the %RDA of one serving of local ethnic food is around 34–46% of the recommended daily intake. This can be better illustrated from one serving of economic noodles (the highest sodium meal), which has 79% of the daily sodium intake advised. From the National Nutrition Survey 2018, it is also evident that Singaporeans are consuming too much salt, with an average daily intake of 9 g [12]. To counter the high sodium foods, high potassium foods can be consumed, as there is abundant evidence that a reduction in dietary sodium and an increase in potassium intake decreases blood pressure and reduces the chances of hypertension, morbidity, and mortality from cardiovascular diseases [19,20].

Calcium is important to prevent osteoporosis and for bone development [21,22]. The recommended dietary intake of calcium for adults should be 1000 mg/day [23]. The consumption of one serving of laska already exceeds the recommended daily level of calcium intake by 124%. In addition, the amount of calcium that can be absorbed varies with an individual’s vitamin D status [24].

Iron insufficiency is the most frequent micronutrient deficit worldwide [25,26]. It is required for many proteins and enzymes, especially hemoglobin to prevent anemia. The recommended dietary intake of iron for adult males aged 18 and above is 8 mg/day. Adult women between ages 18 and 59 require 18 mg/day, while women above 60 require 8 mg/day. In Singapore, one in every two women could be suffering from iron deficiency but they are not aware of it. From the analysis, the average iron content in the local ethnic foods was found to be 1.3 mg/100 g, which is very low. From Table 7, Table 8 and Table 9, we can see that the average %RDA is from 7–29%, with Indian food having a higher average compared to other ethnic groups. Its bioavailability is poor and it is influenced by dietary variables that might either enhance or decrease its availability [27].

## 5. Conclusions

The energy density and the nutrition composition of 45 popular consumed local ethnic cuisines in Singapore were assessed. Indian ethnic cuisine has the highest average energy density of 723 kJ/100 g. This is due to the higher fat and carbohydrate content in Indian cuisine as compared to the Chinese and Malay cuisines. From the ICP-MS analysis, the mineral content of the local ethnic cuisines differs greatly. Overall, Chinese cuisine has the highest amount of sodium in their local ethnic foods. This may be due to the seasonings used to season the food. In addition to the dietary advice and guidelines by the government agencies, the validated data of energy density and macronutrient contents of the local ethnic foods will serve as an important tool in reviewing or setting new dietary guidelines in mitigating health disorders, as well as maintaining sustainable human health in Singapore.

## Figures and Tables

**Figure 1 foods-10-01659-f001:**
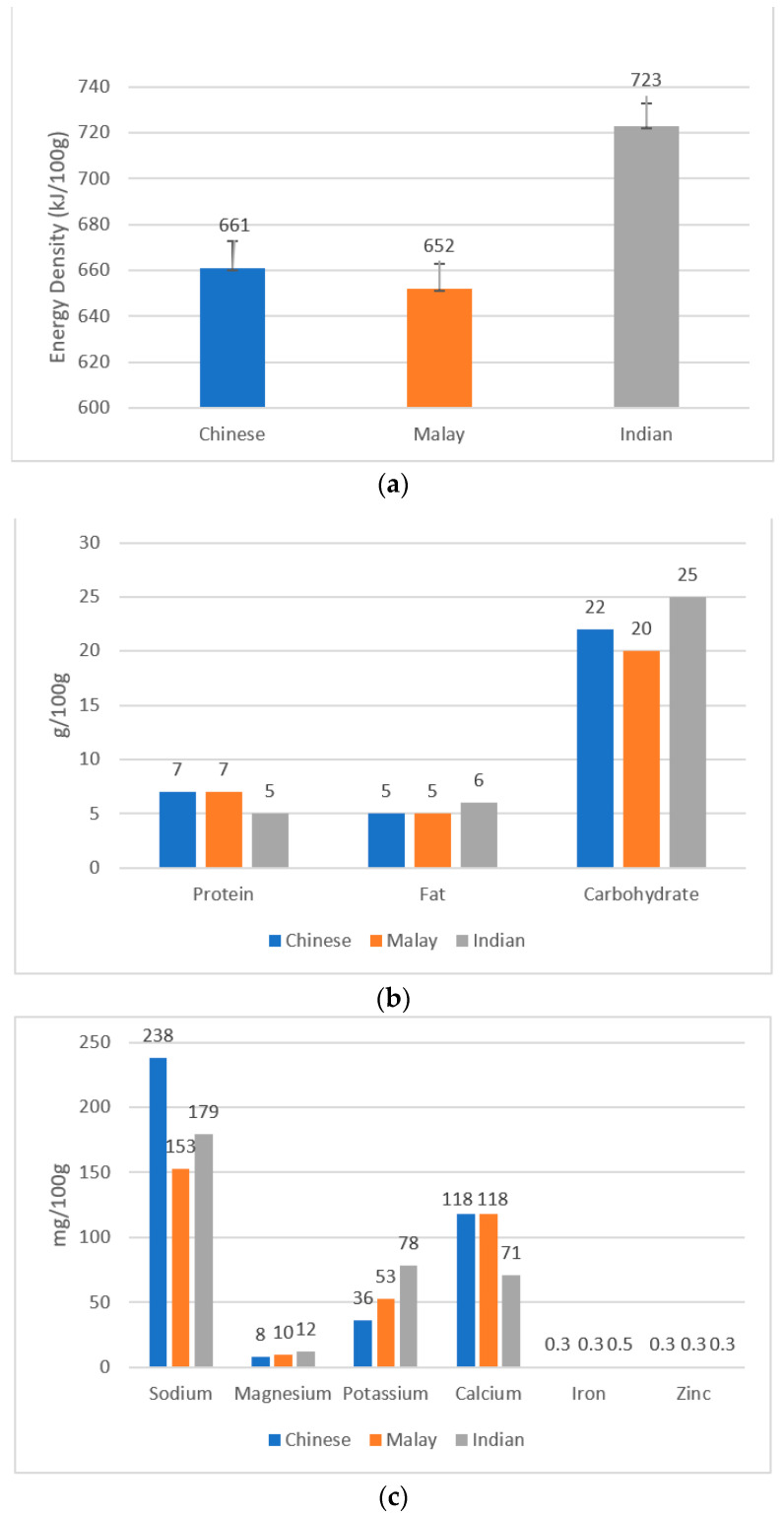
Comparisons of (**a**) energy density, (**b**) macronutrient content, and (**c**) mineral content between the different ethnic cuisines.

**Table 1 foods-10-01659-t001:** Energy and macronutrient content of the commonly consumed Chinese cuisines.

Name of Chinese Local Food	Portion Size (g)	Calories (kJ/100 g)	Protein (g/100 g)	Fat (g/100 g)	Carbohydrate (g/100 g)
Steamed Chicken Rice	412	789 ± 13	9.3 ± 0.1	7.4 ± 0.3	21.2 ± 0.9
Roasted Chicken Rice	285	768 ± 9	8.3 ± 0.2	6.3 ± 0.6	23.4 ± 1.0
Char Kuay Teow	362	762 ± 15	4.0 ± 0.6	6.3 ± 0.2	27.4 ± 0.5
Yang Zhou Fried Rice	388	746 ± 23	5.8 ± 0.2	4.9 ± 0.2	27.7 ± 1.5
Fried Carrot Cake	271	732 ± 8	9.0 ± 0.7	11.5 ± 0.3	8.9 ± 0.1
Sin Chew Bee Hoon	386	689 ± 13	6.6 ± 0.4	5.2 ± 0.1	22.7 ± 0.4
Minced Meat Mee Pok	383	677 ± 5	8.3 ± 0.3	6.9 ± 0.2	16.6 ± 0.3
Economical Mee Goreng	350	677 ± 22	3.5 ± 0.4	2.2 ± 0.2	31.9 ± 1.7
Lor Mai Kai	354	646 ± 12	4.2 ± 0.5	1.7 ± 0.8	30.6 ± 1.5
Laska	577	639 ± 9	8.0 ± 0.4	5.3 ± 0.1	18.3 ± 0.6
Ban Mian Dry	311	621 ± 11	6.7 ± 0.6	4.0 ± 0.2	21.3 ± 1.1
Steamed Chicken Noodle	367	600 ± 7	10.8 ± 0.4	3.3 ± 0.3	17.4 ± 0.7
Fried Hokkien Mee	328	569 ± 18	5.5 ± 0.3	1.5 ± 0.2	25.1 ± 0.7
Char Siew Wanton Noodle	335	502 ± 8	7.9 ± 0.7	1.6 ± 0.2	18.4 ± 0.7
Dumpling You Mian	761	496 ± 15	5.5 ± 0.4	2.6 ± 0.2	18.2 ± 1.4

Values are expressed as mean ± SD. To convert energy to kcal/100 g, divide by 4.184.

**Table 2 foods-10-01659-t002:** Energy and macronutrient content of the commonly consumed Malay cuisines.

Name of Malay Local Food	Portion Size (g)	Calories (kJ/100 g)	Protein (g/100 g)	Fat (g/100 g)	Carbohydrate (g/100 g)
Ayam Penyet	425	849 ± 9	8.9 ± 0.2	9.0 ± 0.1	21.7 ± 0.7
Nasi Kampung Goreng	405	798 ± 11	7.1 ± 0.2	7.1 ± 0.2	24.6 ± 0.8
Nasi Lemak	395	785 ± 10	9.2 ± 0.3	9.0 ± 0.2	17.7 ± 1.0
Ikan Penyet	274	765 ± 14	7.8 ± 0.7	7.0 ± 0.2	22.2 ± 0.8
Nasi Ambang	630	733 ± 5	9.2 ± 0.5	7.2 ± 0.6	18.3 ± 1.7
Goreng Pisang	415	733 ± 2	0.4 ± 0.3	5.6 ± 0.2	30.9 ± 0.2
Tahu Goreng	402	690 ± 9	10.3 ± 0.4	9.9 ± 0.3	8.8 ± 0.9
Mee Soto	281	640 ± 5	9.2 ± 0.3	2.9 ± 0.2	20.3 ± 0.5
Mee Bandung	466	622 ± 7	7.5 ± 0.3	5.2 ± 0.0	18.1 ± 0.2
Mee Bakso	435	611 ± 8	5.7 ± 0.7	2.2 ± 0.1	25.9 ± 0.6
Lotong	269	593 ± 10	8.2 ± 0.3	6.6 ± 0.2	12.4 ± 0.5
Mee Siam	566	590 ± 0	4.9 ± 0.0	2.7 ± 0.0	24.3 ± 0.0
Kentang Ball with Rice Cube	424	490 ± 11	5.4 ± 0.5	3.9 ± 0.3	15.1 ± 0.2
Soto Ayam	693	481 ± 19	6.2 ± 0.8	1.5 ± 0.3	19.1 ± 0.3
Mee Rebus	582	472 ± 5	3.7 ± 0.4	2.4 ± 0.8	19.0 ± 1.4

Values are expressed as mean ± SD. To convert energy to kcal/100 g, divide by 4.184.

**Table 3 foods-10-01659-t003:** Energy and macronutrient content of the commonly consumed Indian cuisines.

Name of Indian Local Food	Portion Size (g)	Calories (kJ/100 g)	Protein (g/100 g)	Fat (g/100 g)	Carbohydrate (g/100 g)
Original Appam	344	1042 ± 11	2.9 ± 0.6	5.4 ± 0.6	47.1 ± 0.6
Egg Appam	304	884 ± 17	4.2 ± 0.4	6.4 ± 0.3	34.7 ± 0.3
Roti Prata	246	869 ± 6	2.5 ± 0.5	6.2 ± 0.3	35.5 ± 0.5
Egg Prata with Chicken Curry	477	782 ± 13	7.7 ± 0.5	7.1 ± 0.5	23.1 ± 1.2
Boneless Mutton Biryani	459	728 ± 8	8.7 ± 0.0	8.9 ± 0.5	14.7 ± 1.1
Poori Set	474	715 ± 13	6.6 ± 0.2	6.8 ± 0.1	21.0 ± 0.9
Marsala Thosai	569	714 ± 13	7.0 ± 0.5	6.9 ± 0.2	20.2 ± 0.3
Naan	597	697 ± 7	5.2 ± 0.8	3.9 ± 0.4	27.5 ± 1.3
Chapatti Set with Potato	336	695 ± 5	6.0 ± 0.6	6.1 ± 0.2	21.8 ± 0.8
Putu Mayam	193	678 ± 4	1.8 ± 0.4	2.6 ± 0.2	32.8 ± 0.4
Chapatti Set with Potato Marsala	603	652 ± 14	4.5 ± 0.4	3.2 ± 0.2	27.2 ± 0.7
Idli Set	389	639 ± 9	6.1 ± 0.1	6.5 ± 0.6	17.4 ± 1.7
Egg Thosai	325	623 ± 8	7.5 ± 0.3	5.7 ± 0.5	17.0 ± 1.1
Vegetable Biryani	640	595 ± 6	4.2 ± 1.1	5.6 ± 0.6	18.8 ± 2.0
Veg Set Meal (Biryani Rice)	513	521 ± 11	3.6 ± 0.5	3.2 ± 0.5	20.3 ± 1.2

Values are expressed as mean ± SD. To convert energy to kcal/100 g, divide by 4.184.

**Table 4 foods-10-01659-t004:** Energy and macronutrient content and the %RDA of the commonly consumed Chinese cuisines.

Name of Local Food	Calories (kcal)	%RDA (Men)	% RDA (Women)	Protein (g)	%RDA (Men)	% RDA (Women)	Fat (g)	%RDA (Men)	% RDA (Women)	Carbohydrates (g)	%RDA (Men)	% RDA (Women)
Steamed Chicken Rice	778	30%	38%	38	50%	61%	30	35%	45%	88	22%	29%
Roasted Chicken Rice	524	20%	26%	24	31%	38%	18	21%	27%	67	17%	22%
Char Kuay Teow	659	25%	32%	15	19%	23%	23	26%	33%	99	25%	32%
Yang Zhou Fried Rice	475	18%	23%	23	30%	36%	19	22%	28%	107	28%	35%
Fried Carrot Cake	475	18%	23%	25	32%	39%	31	36%	46%	24	6%	8%
Sin Chew Bee Hoon	635	24%	31%	26	34%	41%	20	23%	30%	88	23%	29%
Minced Meat Mee Pok	620	24%	30%	32	42%	51%	26	31%	39%	64	16%	21%
Economical Mee Goreng	566	22%	28%	12	16%	20%	8	9%	11%	112	29%	37%
Lor Mai Kai	547	21%	27%	15	19%	24%	6	7%	9%	108	28%	35%
Laska	881	34%	43%	46	60%	73%	30	35%	45%	106	27%	35%
Ban Mian Dry	462	18%	23%	21	27%	34%	13	15%	19%	66	17%	22%
Steamed Chicken Noodle	527	20%	26%	40	52%	64%	12	14%	18%	64	16%	21%
Fried Hokkien Mee	446	17%	22%	18	24%	29%	5	6%	7%	82	21%	27%
Char Siew Wanton Noodle	402	15%	20%	27	35%	42%	5	6%	8%	62	16%	20%
Dumpling You Mian	903	35%	44%	42	55%	67%	20	23%	29%	139	36%	45%
Average	593	23%	29%	27	35%	43%	18	21%	26%	85	22%	28%

**Table 5 foods-10-01659-t005:** Energy and macronutrient content and the %RDA of the commonly consumed Malay cuisines.

Name of Local Food	Calories (kcal)	%RDA (Men)	% RDA (Women)	Protein (g)	%RDA (Men)	% RDA (Women)	Fat (g)	%RDA (Men)	% RDA (Women)	Carbohydrates (g)	%RDA (Men)	% RDA (Women)
Ayam Penyet	862	33%	42%	38	50%	60%	38	44%	56%	92	24%	30%
Nasi Kampung Goreng	772	30%	38%	29	38%	46%	29	33%	42%	100	26%	33%
Nasi Lemak	741	29%	36%	36	47%	58%	35	41%	52%	70	18%	23%
Ikan Penyet	502	19%	25%	21	28%	34%	19	22%	28%	61	16%	20%
Nasi Ambang Chicken	1104	43%	54%	58	76%	93%	46	53%	67%	115	30%	38%
Goreng Pisang	727	28%	36%	2	2%	3%	23	27%	34%	128	33%	42%
Tahu Goreng Set	664	26%	33%	41	54%	66%	40	46%	58%	35	9%	12%
Mee Bandung	693	27%	34%	35	46%	56%	24	28%	36%	84	22%	28%
Mee Bakso	635	24%	31%	25	32%	39%	9	11%	14%	113	29%	37%
Lotong	381	15%	19%	22	29%	35%	18	21%	26%	33	9%	11%
Mee Siam	798	31%	39%	28	36%	44%	15	17%	22%	138	35%	45%
Mee Soto	406	16%	20%	26	34%	41%	8	9%	12%	57	15%	19%
Kentang Ball with Rice Cube	496	19%	24%	23	30%	37%	17	19%	25%	64	16%	21%
Soto Ayam	797	31%	39%	43	56%	68%	10	12%	15%	133	34%	43%
Mee Rebus	655	25%	32%	21	28%	34%	14	16%	21%	111	28%	36%
Average	682	26%	33%	30	39%	48%	23	27%	34%	89	23%	29%

**Table 6 foods-10-01659-t006:** Energy and macronutrient content and the %RDA of the commonly consumed Indian cuisines.

Name of Local Food	Calories (kcal)	%RDA (Men)	% RDA (Women)	Protein (g)	%RDA (Men)	% RDA (Women)	Fat (g)	%RDA (Men)	% RDA (Women)	Carbohydrates (g)	%RDA (Men)	%RDA (Women)
Original Appam	941	36%	46%	27	36%	44%	20	23%	29%	164	42%	54%
Egg Appam	642	25%	31%	12	15%	19%	19	22%	28%	106	27%	35%
Roti Prata	511	20%	25%	6	8%	10%	15	18%	22%	87	22%	29%
Egg Prata	891	34%	44%	37	48%	59%	34	39%	50%	110	28%	36%
Boneless Mutton Biryani	799	31%	39%	40	52%	64%	41	47%	60%	67	17%	22%
Poori Set	811	31%	40%	31	41%	50%	32	37%	47%	100	26%	33%
Marsala Thosai	971	37%	48%	40	52%	63%	39	45%	58%	115	30%	38%
Naan Set	995	38%	49%	31	41%	50%	23	27%	35%	164	42%	54%
Chapati Set with Potato	558	22%	27%	20	27%	32%	21	24%	30%	73	19%	24%
Putu Mayam	312	12%	15%	4	5%	6%	5	6%	7%	63	16%	21%
Chapati Set with Potato Marsala	941	36%	46%	27	36%	44%	20	23%	29%	164	42%	54%
Idli Set	594	23%	29%	24	31%	38%	25	29%	37%	68	17%	22%
Egg Thosai	484	19%	24%	24	32%	39%	19	21%	27%	55	14%	18%
Vegetable Biryani	911	35%	45%	27	35%	43%	36	41%	53%	120	31%	39%
Veg Set Meal	646	25%	32%	19	24%	30%	17	19%	24%	104	27%	34%
Average	734	28%	36%	25	32%	39%	24	28%	36%	104	27%	34%

**Table 7 foods-10-01659-t007:** Mineral content and the %RDA of the commonly consumed Chinese cuisines.

Food	Amount of Na (mg)	% RDA	Amount of Ca (mg)	% RDA	Amount of Fe (mg)	% RDA(Men)	% RDA(Women)
Steamed Chicken Rice	638	32%	1127	113%	1.65	21%	9%
Roasted Chicken Rice	489	24%	930	93%	0.28	4%	2%
Char Kuay Teow	929	46%	408	41%	2.17	27%	12%
Yang Zhou Fried Rice	1046	52%	172	17%	0.78	10%	4%
Fried Carrot Cake	427	21%	245	25%	1.63	20%	9%
Sin Chew Bee Hoon	932	47%	541	54%	1.54	19%	9%
Minced Meat Mee Pok	1041	52%	184	18%	1.53	19%	9%
Economical Mee Goreng	1575	79%	250	25%	0.7	9%	4%
Lor Mai Kai	1154	58%	170	17%	0.71	9%	4%
Laska	1380	69%	1236	124%	2.89	36%	16%
Ban Mian Dry	755	38%	267	27%	0.62	8%	3%
Steamed Chicken Noodle	625	31%	284	28%	0.73	9%	4%
Fried Hokkien Mee	861	43%	207	21%	0.98	12%	5%
Char Siew Wanton Noodle	731	37%	212	21%	0.67	8%	4%
Dumpling You Mian	1086	54%	812	81%	1.52	19%	8%
Average	911	46%	470	47%	1	15%	7%

**Table 8 foods-10-01659-t008:** Mineral content and the %RDA of the commonly consumed Malay cuisines.

Food	Amount of Na (mg)	% RDA	Amount of Ca (mg)	% RDA	Amount of Fe (mg)	% RDA (Men)	%RDA (Women)
Ayam Penyet	415	21%	352	35%	1.27	16%	7%
Nasi Kampung Goreng	746	37%	272	27%	1.21	15%	7%
Nasi Lemak	459	23%	459	46%	1.97	25%	11%
Ikan Penyet	390	20%	908	91%	0.55	7%	3%
Nasi Ambang Chicken	919	46%	1002	100%	3.78	47%	21%
Goreng Pisang	498	25%	361	36%	0.83	10%	5%
Tahu Goreng Set	573	29%	1129	113%	2.01	25%	11%
Mee Bandung	625	31%	329	33%	1.4	18%	8%
Mee Bakso	734	37%	234	23%	0.87	11%	5%
Lotong	312	16%	403	40%	0.81	10%	5%
Mee Siam	1156	58%	399	40%	2.26	28%	13%
Mee Soto	798	40%	159	16%	0.84	11%	5%
Kentang Ball with Rice Cube	491	25%	531	53%	2.12	27%	12%
Soto Ayam	811	41%	315	32%	1.39	17%	8%
Mee Rebus	1170	59%	453	45%	1.16	15%	6%
Average	673	34%	487	49%	1	19%	8%

**Table 9 foods-10-01659-t009:** Mineral content and the %RDA of the commonly consumed Indian cuisines.

Food	Amount of Na (mg)	%RDA	Amount of Ca (mg)	% RDA	Amount of Fe (mg)	% RDA (Men)	% RDA (Women)
Original Appam	340	17%	77.5	8%	1.03	13%	6%
Egg Appam	254	13%	200	20%	1.52	19%	8%
Roti Prata	617	31%	179	18%	0.74	9%	4%
Egg Prata	1057	53%	516	52%	2.86	36%	16%
Boneless Mutton Biryani	868	43%	368	37%	1.84	23%	10%
Poori Set	877	44%	399	40%	5.21	65%	29%
Marsala Thosai	999	50%	316	32%	3.41	43%	19%
Naan Set	1219	61%	537	54%	2.99	37%	17%
Chapati Set with Potato	862	43%	275	28%	2.02	25%	11%
Putu Mayam	185	9%	23.5	2%	0.77	10%	4%
Chapati Set with Potato Marsala	1043	52%	395	40%	4.83	60%	27%
Idli Set	408	20%	186	19%	2.33	29%	13%
Egg Thosai	888	44%	219	22%	1.62	20%	9%
Vegetable Biryani	1100	55%	677	68%	1.92	24%	11%
Veg Set Meal	1021	51%	287	29%	1.54	19%	9%
Average	783	39%	310	31%	2	29%	13%

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
