# Peer review of "Energy Content and Nutrient Profiles of Frequently Consumed Meals in Singapore"

_foods, 2021, doi:10.3390/foods10071659_

Round 1

Reviewer 1 Report

The article is clear and as indicate in the objective it is presented the nutritional composition of 15 meals from three ethnic groups in Singapor (Chinese, Malay and Indian). The results are well presented however it is necessary more information related the ingredients and food that are included in each meal, and the other hand the frequency of the consumption and a discusion related to the portion size, because it is the main reason to correlate with obesity or other diet disease (diabetis, hypertension, cholesterol). Other important point it is the lipidic profile that has not been analyzed or not included, because nowadays it is the main factor in lipidemic diseases. The figures 2, 3, 4, 5, are to high and it shown the mineral composition in this sense this information has to be included as supplementary information. And to finally all the results have to be compared with guidance recomendations from institutional organism or international associations (FDA,EFSA....) or health institutions, just to compare and evaluate the diet balance.

Author Response

Thanks.

Reviewer 2 Report

Very interesting work. I would suggest performing statistics with regard to the macro and micronutrients of different foods. This will allow evaluating if differences between meals/nutrients are significant. Please find suggestions in the attached file.

Author Response

Thanks.

Reviewer 3 Report

  1. The quantification methods are sound, and the measurements/data are sufficient. However, it may be better to present all nutrient measurements in relative terms (%RV), for example, each meal per serving would provide what percentage of daily recommended intakes (RDIs) of iron, zinc, protein, etc. It is not easy for general consumers/readers (if not trained nutritionists) to comprehend all absolute values (measurements) directly.
  2. Were all meal samples obtained from the same location? Are they representative for the same type across other locations?
  3. What is the rationale/reasoning to select the specific 15 meals from each of the three ethnic groups? Are these 15 meals mostly consumed by the majority of the local populations?
  4. How are the nutrient measurements in comparison with the national dietary guideline? Or, what genuine implications can be extrapolated from the huge variations of these many meals? 

Author Response

Thanks.
